# Recent Advances in District Cooling Diffusion in the EU27+UK: An Assessment of the Market

Simon Pezzutto [1,*], Philippe Riviere [2,3], Lukas Kranzl [4], Andrea Zambito [1], Giulio Quaglini [1], Antonio Novelli [5], Marcus Hummel [6], Luigi Bottecchia [1] and Eric Wilczynski [1]

1   Institute for Renewable Energy, European Academy of Bolzano (EURAC Research), Viale Druso 1, 39100 Bolzano, Italy; andrea.zambito@eurac.edu (A.Z.); giulio.quaglini@eurac.edu (G.Q.); luigi.bottecchia@eurac.edu (L.B.); eric.wilczynski@eurac.edu (E.W.)
2   Directorate-General for Energy (DG Energy), European Commission, Unit B3: Buildings and Products, 1049 Bruxelles, Belgium; philippe.riviere@ec.europa.eu or philippe.riviere@mines-paristech.fr
3   Center for Energy-efficient Systems (CES), Department of Energy and Processes (DEP), Mines ParisTech, PSL University, 60 Boulevard Saint-Michel, CEDEX 06, 75272 Paris, France
4   Energy Economics Group, Institute of Energy Systems and Electric Drives, TU Wien, Gusshausstrasse 25-29/370-3, 1040 Vienna, Austria; kranzl@eeg.tuwien.ac.at
5   Planetek Italia, Via Massaua 12, I-70132 Bari, Italy; antonio.novelli87@gmail.com
6   E-Think Energy Research, Argentinierstrasse 18/10, 1040 Vienna, Austria; hummel@e-think.ac.at
*   Correspondence: simon.pezzutto@eurac.edu; Tel.: +39-0471-055-622

**Abstract:** The scope of this investigation is to analyze recent advances in district cooling (DC) in Europe (European Union plus the United Kingdom (EU27+UK)). The study focuses on data and information from the past decade and draws a picture for the year 2016. To date, in contrast to the European district heating market, the European DC branch has barely been explored in scientific literature. In the current paper, the data that describe the actual DC market in the EU27+UK—which includes the quantity of DC plants, equivalent full-load hours, and installed capacities as well as the values for seasonal energy efficiency—are collected and then explored using a bottom-up approach. The results indicate that DC is responsible for a minor part of the useful energy demand of Europe for cooling with around 3 TWh/y. Overall, the share of the useful energy demand for DC corresponds with 1 to 2% of the EU27+UK useful energy demand. Moreover, it is worth mentioning that the penetration of DC varies considerably from country to country, and eight European Member States appear to not have district cooling systems at all. Lastly, it is observed that DC has been slowly but constantly growing in Europe for decades and is characterized by a high growth potential, particularly in the service sector.

**Keywords:** district cooling; Europe; diffusion; market; potential

## 1. Introduction

The European Union (EU) has taken numerous steps to minimize its carbon footprint. The European Member States (MSs) have made commitments to cut greenhouse gas (GHG) emissions by at least 20% versus 1990 levels, increase the supply of energy produced by renewable energy sources (RESs) to at least 20% compared with 2020 levels, and reduce primary energy consumption by 20% through improvements in energy efficiency [1].

By 2030, the EU plans to further reduce domestic GHG emissions by at least 55% compared with 1990 levels. This target is proposed to be enhanced to 55%, according to the European Green Deal announced in September 2020 [2]. EU member states will use a coordinated effort to provide sound regulatory frameworks to investors with an integrated policy framework. National policies designed to facilitate permanent energy efficiency improvements have been designed to increase RES energy production to 32% [3].

European Union MSs have announced their intention to lower European GHG emissions by 80–95% versus 1990 levels by 2050. Fulfilling the Paris Conference of Parties 21

(COP21) agreement and the European Green Deal requires a further reduction in GHG emissions [4,5]. As developed nations are anticipated to reduce GHG emissions earlier than developing nations, EU MSs will need to reevaluate their energy and climate policy targets [6].

In 2018, primary energy utilization in the EU27+UK was approximately 1600 Mtoe/y [7]. Heating and cooling (H&C) functions constitute the principal sources of energy consumption; building space cooling (SC), space heating (SH), domestic hot water (DHW), and industrial heat comprise about 800 Mtoe/y. Consumption from electricity and transportation accounts for the other 800 Mtoe/y, providing 310 and 490 Mtoe/y, respectively [8].

There have been considerable investments in recent years by EU MSs to quantify energy use in different sectors [9–12]. However, as opposed to energy use data for the SH and DHW sectors, there is only a limited amount of data available for consumption in the SC sector, especially concerning DC [8,12]. It is worth noting that district cooling systems (DCSs) are characterized by relevant energy savings for SC as well as process cooling (PC) applications [13–15]. Notably, the efficiency of DC systems can be up to ten times higher than stand-alone on-site distributed solutions [16,17].

Cooling refers to the removal of heat [18]. In the following text, the term cooling is intended as SC and PC whereas it is specified when referring to one of the two cooling types indicated above.

Space cooling is defined as the removal of heat from the air to cool indoor air and ensure healthy conditions and thermal comfort for the occupants of an enclosed space (e.g., buildings). Thus, SC lowers the temperature of the air. Typical set-points of indoor air temperature for SC vary, occurring between 20 and 30 °C [19]. In contrast, PC is defined as the removal of heat from processes (e.g., plastic mold cooling [20]), from products, or from a confined space containing these processes or products) to maintain the required set temperature. District cooling refers to the dissemination of cold thermal energy in the form of chilled liquids from central or decentralized sources of production to a network and multiple buildings or sites for the use of SC and/or PC [21]. DC provides SC and PC primarily for commercial and public buildings, but also for the industrial and residential sectors [13].

It has been observed that, in recent years, factors such as climate change driven by rising heat and moisture during summers as well as the rise of welfare throughout the European population, developing building design, and mounting predilection for thermal comfort have recently led to an increase in cooling demand.

The study team experienced notable difficulties in finding data/information regarding the investigated field. In contrast to the EU district heating (DH) market, the European DC branch has been barely explored in scientific literature so far. There is a real lack of data and information. Data and information regarding the treated topic are often difficult to find, not freely accessible, outdated, or mixed in different repositories as well as fragmented.

This work aims to produce a high-quality dataset on the DC market with an emphasis on completeness, accuracy, and reliability. In particular, a special focus in the framework of our analysis has been placed on the following features:

- Data inventory;
- Data reliability;
- Data definition and comparability.

### 1.1. Data Inventory

A major obstacle that arises when creating an inventory of DC plants and respective technical characteristics in the EU27+UK involves the preparation of a comprehensive list of all existing data. In general, data that have been gathered at the European level present exceptional benefits due to their broad geographical scope (e.g., Euroheat & Power [22]). Nevertheless, data completeness can never be completely guaranteed. Efforts to close data gaps require the extrapolation and assembly of data from large online data sources (e.g., the Intelligent Energy Europe (IEE) project RESCUE (Renewable Smart Cooling for

Urban Europe) [16]). Searching for data source-by-source is required to assure a thorough approach in addressing data gaps, especially through the use of individual scientific literature sources, including journal papers (e.g., Buffa et al. [23], Inayat et al. [24], Eveloy et al. [25], and Gang et al. [26]), books (e.g., Colmenar-Santos et al. [27]), project reports (e.g., Dittmann et al. [28]), and conference proceedings (e.g., Truelsen and Forsyning [29]). Only through such an in-depth approach can the aforementioned data gaps be filled. The data collection methodology is described in Section 2.

### 1.2. Data Reliability

Significant efforts were devoted to analyzing sources, assessing the dependability of the collected data, and filling current gaps with comprehensive studies. Several types of data were examined through the analysis of different methods applied to collecting the identified data (e.g., the amount of DC plants versus DC sales). Data with insufficient or ambiguous documentation were not considered when developing the dataset. All of the information gathered on DC (i.e., the number of plants in operation, the number of installed capacities, energy efficiency values, and the equivalent full-load hours per country) were statistically cleaned and assessed (the employed methodology is detailed in Section 2). Additionally, supplementary sources and types of information were used to verify the results acquired for the EU27+UK (see Section 4).

### 1.3. Data Definition and Comparability

Despite the use of standardized data formats and units by many data suppliers, one cannot automatically imply that data are completely comparable. To improve the comparability of the data, the whole data elaboration procedure requires the adjustment of discrepancies that result from different measures, methods, specifications, assumptions, and time references [30].

Data were gathered for all MSs from the most current records; data that were at least ten years old were not included. The work that has been carried out, including the documentation development, is anticipated to enhance the data quality, increase the value of the existing data, and provide the necessary data for monitoring the development of the DC field in the EU.

To obtain a complete picture of DC in Europe, the cooling part of district heating and cooling (DHC) plants all over the EU27+UK is also taken into consideration within the present investigation.

When assessing the data comparability, the terms "energy demand" and "energy consumption" are often mistakenly used interchangeably in scientific literature. In this study, we distinguish between useful energy demand (UED) and final energy consumption (FEC) for cooling. The UED is the net heat removed from the space/process to be cooled. In contrast, the FEC for cooling is the energy input of the DC plant. As a result, the respective quantities differ by disparate conversion factors. The energy efficiency ratio (EER) for electrically driven cooling equipment is >1. Due to that, the FEC for cooling is lower than the UED for cooling [14].

The following text concerns the DC final energy consumption/useful energy demand in the EU27+UK, country-by-country, and provides an overview concerning the diffusion of the respective technology. The reference year of the following results is 2016.

The main findings indicate DC to be responsible for a limited portion of the useful cooling demand of Europe (about 3 TWh/y), which equals about 1 to 2% of the total cooling market of Europe. The presence of DCSs varies considerably from country to country, and eight EU MSs did not have DC at all. DC has been slowly but gradually growing in Europe for decades as is characterized by a high potential for growth, especially in service sector buildings.

Within the following text, when referring to the final DC consumption we intend the final energy consumption of DC plants to deliver cooling to customers whereas useful DC demand refers to DC sales.

Section 1 presents the research, positioning it in an extensive context. Section 2 details the data and information gathering as well as the applied methodology. Section 3 specifies the main outcomes. Section 4 provides the discussion. Lastly, Section 5 indicates the conclusions.

## 2. Materials and Methods

For this study, the key sources of data collection were concentrated on prior investigations. Specifically, Euroheat & Power has been involved in various investigations and projects to analyze the topic of DC. The latter includes several so-called "Country-by-Country" statistics, carried out every two years and dating back to 2013 [26,31–33], as well as the DHCities initiative [34], IEE and Horizon 2020 (H2020) projects, and STRATEGO: Multi-level actions for enhanced Heating & Cooling plans and Heat Roadmap Europe 4 (HRE4) [35,36].

The International Energy Agency (IEA) is also active in the investigated field, carrying out many projects on DHC [37] and performing the International Symposiums on District Heating and Cooling (DHC), which have covered the DC aspect since 1999 [38], as well as producing several reports, entailing highly valuable data/information on the investigated field (e.g., [39]).

A crucial data/information repository is provided by Halmstad University, which created a unique database on the investigated topic (the Halmstad University District Heating and Cooling Database (HUDHC)) [40].

Other important information sources were also used such as papers from academic publications (e.g., Passerini et al. [41], Inayat et al. [24], Eveloy et al. [25], and Gang et al. [26]), PhD theses (e.g., Persson [42]), conference proceedings (e.g., Casetta [43]), project deliverables (e.g., Tvärne et al. [44]), and books (e.g., Frederiksen and Werner [45]). Notably, according to Inayat et al., a DCS results in high-level energy efficiency compared with standard cooling types of machinery [24]. Moreover, according to Gang et al., the integration of DCSs with RES can develop energy efficiency. Renewable energy sources such as biomass, solar-thermal, geothermal, solar-photovoltaic, and waste heat are the best fitting RES for DCSs.

It is worth mentioning that although the previously noted sources provide data/information on the location of DC plants and related technical specificities, a clear picture of the DC market in Europe is currently missing.

To assess DC final energy consumption/useful energy demand (DC sales) for cooling purposes per country (EU27+UK), we followed the methodology provided by [46].

First, the authors mainly researched the following data per country:

- Number of DC plants;
- Equivalent full-load hours (EFLHs);
- Capacities installed;
- Energy efficiency levels (seasonal energy efficiency ratio (SEER)).

The work input of the DC plants per country was calculated. To acquire these values, the total DC capacity of the countries was divided by their respective SEER means. Notably, the acronym SEER stands for seasonal energy efficiency ratio, which is the proportion of the cooling productivity of a DC plant over a conventional cooling season distributed by the energy it employs in watts/hours [47]. Although the energy efficiency ratio (EER) is computed using rated load requirements, the SEER was chosen because it signifies the actual usage ratings of DC plants for the cooling season over the course of a year [47–49]. In Equation (1) below, SEER demonstrates the total heat quantity that is removed over the whole annual cooling season ($Q_{cold,season}$) divided by the total work input (neglecting DC powered by sorption equipment) of the cooling equipment for the duration of the same period ($W_{season}$) [48]:

$$SEER = \frac{Q_{cold,\ season}}{W_{season}} \tag{1}$$

An extensive analysis of the literature was performed to calculate the dependable values using the specified bottom-up methodology. Only sources from academic literature were used for the data collection. All gathered information was filtered and statistically evaluated. With regard to the number of permitted sources, data that were plus or minus one standard deviation beyond the range around the average of the respective data pool were discarded. The filtered values were used to calculate a more robust average. Unfortunately, assembling two or more data per researched value was not always possible, so, in these cases, no statistical elaboration was performed.

Conclusively, the FEC by DC plants and country was calculated. To acquire the annual final DC consumption per country, the average equivalent full-load hours (time, T) within a year were multiplied with the respective work input (W):

$$\text{FEC}_{\text{cooling}} = \text{T}_{\text{equivalent full-load hours}} \times \text{W} \qquad (2)$$

The equivalent full-load hours (EFLHs) were obtained by dividing the DC sales (TWh/y) by the respective capacity installed (MW) per country found mainly in the source [22]. Specifically, EFLHs define the intentional degree of use of a technical system. The term is mostly employed in the field of energy manufacture plants by describing the period for which DC machinery is required to function at its nominal power (capacity, kW) to transfer the same quantity of work in a specific time [24]. In this study, the word was suggested to be the yearly number of hours in which the DCS was fully operational.

We did not differentiate among FECs between different sectors because the sources utilized did not allow the performing of such calculations for the entirety of EU27+UK.

The energy efficiency parameter utilized as the input for Equation (2)—the SEER—was provided by Deliverable 3.2 of the Heat Roadmap Project (HRE) [36].

Furthermore, to obtain the following UED for DC (DC sales), the installed capacity ($\text{C}_{\text{installed}}$) was multiplied by the EFLHs ($\text{T}_{\text{equivalent full-load hours}}$) per country, but not taking into consideration the SEER:

$$\text{UED}_{\text{district cooling}} = \text{T}_{\text{equivalent full-load hours}} \times \text{C}_{\text{installed}}. \qquad (3)$$

In Section 3, we provide indications regarding the portion of free cooling (in a percentage) of the energy supplied by DC plants in various countries (EU27+UK) subdivided into Temperate, Warm, and Cold Europe. The latter data could be found solely by country group and were the results of modeling provided by Deliverable 3.2 of the Heat Roadmap project (HRE) [36]. This also applied to the SEER values utilized to perform the calculations exposed in Section 3.

The map given in Figure 1, which shows the distribution of DC plants in Europe, was created by georeferencing the collected data using geographic information systems (GIS); specifically, QGIS 3.6 [50].

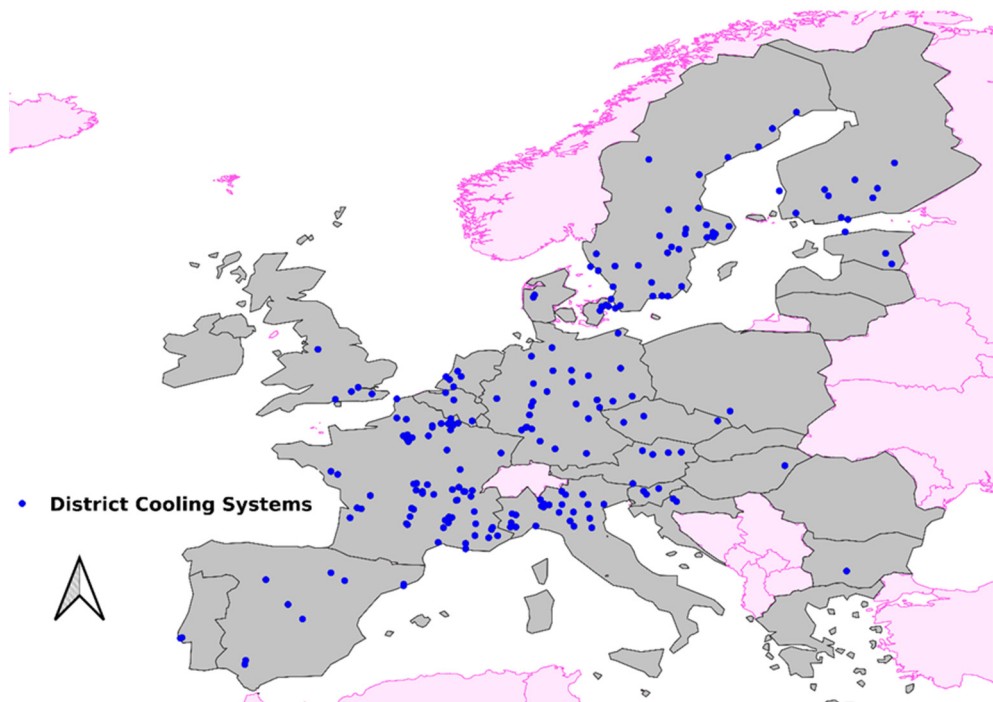

**Figure 1.** Diffusion of district cooling systems per country, EU27+UK, reference year 2016 [17,26–29,34,44,51–54] (blue dots on the map indicate the location of district cooling systems).

## 3. Results

Figure 1 provides indications about the diffusion of DCSs in the EU27+UK. In total, more than 200 applications were identified.

As visible in Figure 1, the majority of DC applications could be found in France, Sweden, Germany, and Italy followed by Finland and Spain with the same number of installations. The remaining countries were characterized by a minor number of plants.

It has to be underlined that the Nordic countries of the EU (Denmark, Finland, and Sweden) were characterized by a relatively high number of installations despite cold weather conditions. Possible reasons for that could be the use of cooling provided by DC plants for PC applications, a higher sensibility of Nordic populations regarding ambient temperature increases, a higher tradition of a grid-connected heating and cooling supply, and buildings optimized to reduce heating consumption in winter conditions (high insulation levels), which leads to a rise of SC use as well as a higher presence of DH networks that facilitate the implementation of DCSs.

Table 1 provides indications regarding the capacity installed per country. Please note that countries marked with a slash "/" refer to countries for which the indicated sources mention that there was no DCS present at that time.

**Table 1.** District cooling capacity installed per country (MW) and respective percentage distribution, EU27+UK, reference year 2016 [17,26,28,41,43,45,47,54–57].

| Countries | Capacity Installed (MW) | Percentage |
| --- | --- | --- |
| Austria | 130 | 1.66 |
| Belgium | 1 | 0.01 |
| Bulgaria | 0.5 | 0.01 |
| Croatia | 7 | 0.09 |
| Cyprus | / | / |
| Czech Republic | 35 | 0.45 |
| Denmark | 22 | 0.28 |
| Estonia | 13 | 0.17 |
| Finland | 283 | 3.62 |

**Table 1.** *Cont.*

| Countries | Capacity Installed (MW) | Percentage |
|---|---|---|
| France | 761 | 9.74 |
| Germany | 241 | 3.09 |
| Greece | / | / |
| Hungary | 1 | 0.01 |
| Ireland | / | / |
| Italy | 202 | 2.59 |
| Latvia | / | / |
| Lithuania | / | / |
| Luxembourg | 19 | 0.24 |
| Malta | / | / |
| Netherlands | 23 | 0.29 |
| Poland | 43 | 0.55 |
| Portugal | 40 | 0.51 |
| Romania | / | / |
| Slovakia | / | / |
| Slovenia | 5 | 0.06 |
| Spain | 122 | 1.56 |
| Sweden | 5787 | 74.08 |
| United Kingdom | 76 | 0.97 |
| Sum | 7811.5 | 100 |

The amount of DC capacity installed in the base year 2016 totaled more than 7800 MW. As visible in the table above, Sweden was ranked first with nearly 6000 MW, which accounted for almost three-quarters of the entire DC capacity installed in Europe. France followed with almost 800 MW. Finland, Germany, and Italy followed with capacities ranging from about 200 to 300 MW. Austria, Spain, and the UK were next with installed capacities of around 100 MW. The remaining countries were characterized by minor values.

Table 2 displays the indications concerning the SEER of DC plants per country, grouped as Temperate, Warm, and Cold Europe.

**Table 2.** District cooling seasonal energy efficiency ratio per country/country group (Temperate, Warm, and Cold Europe), EU27+UK, reference year 2016 [36] (France is divided into two parts, France North and South, for respective calculations. A mean value among Temperate and Warm Europe is applied for the latter country).

| |
|---|
| Temperate Europe: Austria, Belgium, Czech Republic, France (North), Germany, Hungary, Luxemburg, Netherlands, Poland, Slovakia, Slovenia |
| SEER: 5.1 |
| Warm Europe: Bulgaria, Croatia, Cyprus, France (South), Greece, Italy, Malta, Portugal, Romania, Spain |
| SEER: 4.0 |
| Cold Europe: Denmark, Estonia, Finland, Ireland, Latvia, Lithuania, Sweden, UK |
| SEER: 9.4 |

Figure 2 displays the distribution of DC EFLHs among the EU27+UK countries.

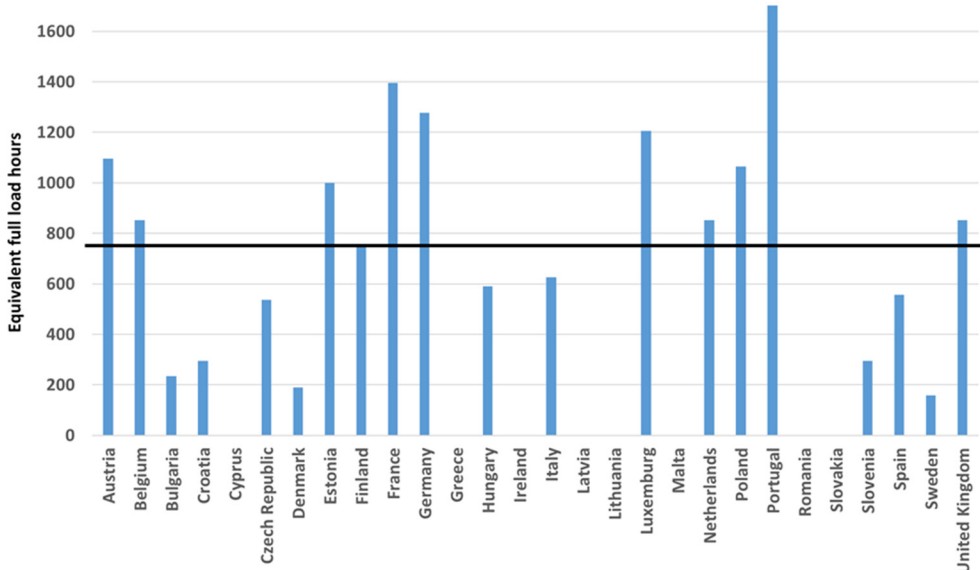

**Figure 2.** District cooling equivalent full-load hours per country, EU27+UK, reference year 2016 [26,28,53–59] (no district cooling systems could be identified in Cyprus, Greece, Ireland, Latvia, Lithuania, Malta, Romania, and Slovakia; respective equivalent full-load hours are not available).

As visible in Figure 2, the DC EFLHs in the EU27+UK reached a maximum value of more than 1600 in Portugal. The minimum value was about ten times less with nearly 160 EFLHs in Sweden. The mean EFLHs for DC in Europe was approximately 780 (please see respective horizontal black line in Figure 2).

The EFLHs, as presented in Figure 2, showed considerable deviations between countries. Countries with similar climates such as Sweden and Finland or Portugal and Spain also showed highly deviating values of EFLHs. There are several reasons, which may help to substantiate this result:

(i) First, the type of consumer connected makes a significant difference (e.g., in a cold climate from about 7500 EFLHs for process cooling to only a hundred EFLHs for residential customers [47,59]).

(ii) Second, the construction of the DC capacity may be ahead of customer connection or the other way round; large customers may disconnect if they find a cheaper supply or the industry closes, etc.

(iii) Third, the moderate number of DC grids for most countries has a substantial potential impact on the first two arguments on the overall result.

Overall, this shows that the above-indicated values of EFLHs should not be considered as representative information of DC or delivery of a starting point for future DC grids to be constructed. Rather, they should be understood as a snapshot of the state of DC in the year 2016.

Integrating the data of Tables 1 and 2, and Figure 2 in Equation (2) resulted in Figure 3.

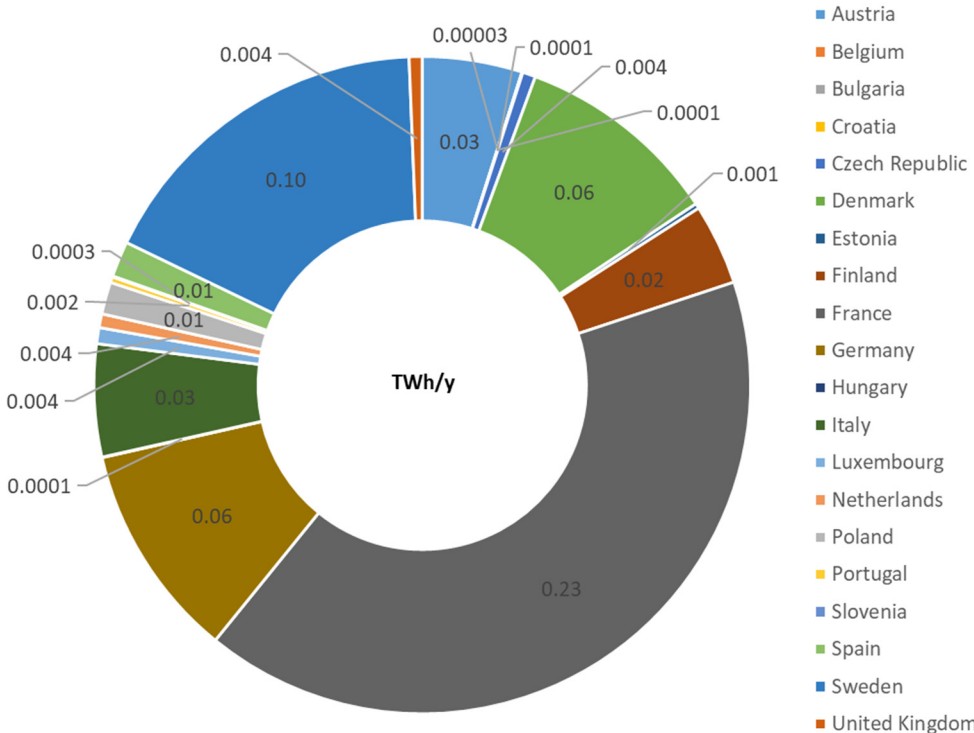

**Figure 3.** District cooling final energy consumption per country, EU27+UK, reference year 2016 [16,28,30,42,43,45,46,53–58].

In the pie chart above (Figure 3), not all obtained numbers per country were easily recognizable. Table 3 also displays the percentage distribution of DC FECs in Europe.

**Table 3.** District cooling final energy consumption per country and respective percentage distribution, EU27+UK, reference year 2016 [16,28,30,42,43,45,46,53–58].

| Countries | TWh/y per Country, Residential, Service, and Industrial Sectors | Percentage |
|---|---|---|
| Austria | 0.03 | 4.91 |
| Belgium | 0.0001 | 0.02 |
| Bulgaria | 0.00003 | 0.005 |
| Croatia | 0.0001 | 0.03 |
| Cyprus | / | / |
| Czech Republic | 0.004 | 0.64 |
| Denmark | 0.06 | 10.10 |
| Estonia | 0.001 | 0.24 |
| Finland | 0.02 | 3.98 |
| France | 0.23 | 40.93 |
| Germany | 0.06 | 10.61 |
| Greece | / | / |
| Hungary | 0.0001 | 0.02 |
| Ireland | / | / |
| Italy | 0.03 | 5.54 |
| Latvia | / | / |
| Lithuania | / | / |
| Luxembourg | 0.004 | 0.79 |
| Malta | / | / |
| Netherlands | 0.004 | 0.68 |
| Poland | 0.01 | 1.58 |
| Portugal | 0.002 | 0.28 |
| Romania | / | / |
| Slovakia | / | / |

**Table 3.** *Cont.*

| Countries | TWh/y per Country, Residential, Service, and Industrial Sectors | Percentage |
|---|---|---|
| Slovenia | 0.0003 | 0.05 |
| Spain | 0.01 | 1.75 |
| Sweden | 0.10 | 17.21 |
| United Kingdom | 0.004 | 0.64 |
| Sum | 0.57 | 100 |

As visible in Figure 3 and Table 3, France was ranked first with more than 0.20 TWh/y. Sweden followed with around 0.10 TWh/y. Germany was next with about 0.06 TWh/y. Italy and Austria followed with about 0.03 TWh/y each. Finland was next with approximately 0.02 TWh/y. Spain and Poland followed with approximately 0.01 TWh/y. The remaining countries showed minor values. The total amount of final DC consumption in Europe reached nearly 0.6 TWh/y.

Integrating the data of Table 1 and Figure 2 in Equation (3) resulted in Figure 4, which visualizes the useful DC cooling demand (DC sales) shares per country.

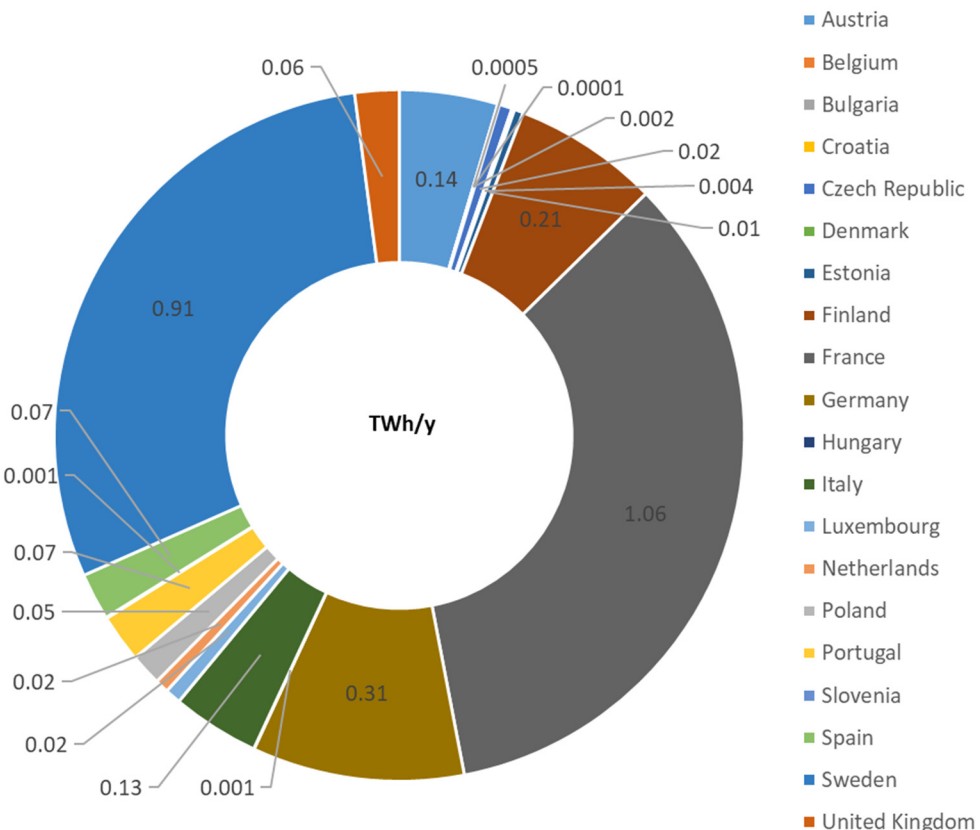

**Figure 4.** Useful district cooling demand (district cooling sales) per country, EU27+UK, reference year 2016 [16,28,30,42,43,45,46,53–58].

In the pie chart above (Figure 4), not all obtained numbers per country were easily recognizable. Table 4 also displays the percentage distribution of useful DC demand in Europe.

**Table 4.** Useful district cooling demand (district cooling sales) per country and respective percentage distribution, EU27+UK, reference year 2016 [16,28,30,42,43,45,46,53–58].

| Countries | TWh/y per Country, Useful Energy Demand | Percentage |
|---|---|---|
| Austria | 0.14 | 4.60 |
| Belgium | 0.0005 | 0.02 |
| Bulgaria | 0.0001 | 0.003 |
| Croatia | 0.002 | 0.07 |
| Cyprus | / | / |
| Czech Republic | 0.02 | 0.60 |
| Denmark | 0.004 | 0.13 |
| Estonia | 0.01 | 0.42 |
| Finland | 0.21 | 6.83 |
| France | 1.06 | 34.31 |
| Germany | 0.31 | 9.94 |
| Greece | / | / |
| Hungary | 0.001 | 0.02 |
| Ireland | / | / |
| Italy | 0.13 | 4.09 |
| Latvia | / | / |
| Lithuania | / | / |
| Luxembourg | 0.023 | 0.74 |
| Malta | / | / |
| Netherlands | 0.02 | 0.63 |
| Poland | 0.05 | 1.48 |
| Portugal | 0.07 | 2.24 |
| Romania | / | / |
| Slovakia | / | / |
| Slovenia | 0.001 | 0.05 |
| Spain | 0.07 | 2.20 |
| Sweden | 0.91 | 29.55 |
| United Kingdom | 0.06 | 2.08 |
| Sum | 3.09 | 100 |

As visible in Figure 4 and Table 4, France was ranked first with more than 1 TWh/y. Sweden followed with almost 1 TWh/y. Germany was next with slightly more than 0.30 TWh/y. Finland followed with more than 0.20 TWh/y. Austria and Italy were next with more than 0.10 TWh/y. Portugal, Spain, and the UK followed with less than 0.1 TWh/y each. The remaining European countries showed minor values. The total amount of useful DC demand (DC sales) in the EU27+UK reached more than 3 TWh/y.

Table 5 indicates the portion of free cooling (in a percentage) of the energy supplied by DCSs in various countries subdivided into Temperate, Warm, and Cold Europe. The latter was collected and elaborated on from the study of Fleiter et al. [60].

**Table 5.** Free cooling portion of district cooling systems per country, Temperate, Warm, and Cold Europe, EU27+UK, reference year 2016 [36,61].

| Temperate Europe: Austria, Belgium, Czech Republic, France (North), Germany, Hungary, Luxemburg, Netherlands, Poland, Slovakia, Slovenia |
|---|
| Free cooling portion: 40% |
| Warm Europe: Bulgaria, Croatia, Cyprus, France (South), Greece, Italy, Malta, Portugal, Romania, Spain |
| Free cooling portion: 20% |
| Cold Europe: Denmark, Estonia, Finland, Ireland, Latvia, Lithuania, Sweden, UK |
| Free cooling portion: 80% |

## 4. Discussion

According to this study, DC was responsible for a minor part of the useful cooling demand of Europe (approximately 3 TWh/y). Overall, the latter corresponded with approximately 1 to 2% of the European cooling market energy demand [19,21,23,60,61].

It was observed that the estimations closely matched the useful DC demand (DC sales) values found in the present study. For instance, the DHC+ Technology Platform indicated that the European DC market energy demand was about 3 TWh/y in 2012 [17]. Tvärne et al. [44] approximated that the EU useful DC energy demand would reach a value of around 3 TWh/y. Gudmundsson et al. also indicated a value of approximately 3 TWh/y [54]. In contrast, the useful energy demand for DC calculated in this study slightly exceeded the findings from Dalin et al. [13]. The latter-mentioned work estimated it to be about 2.3 TWh/y. However, taking into consideration that the above-mentioned indication referred to the year 2003, the provided outcome appeared to be similar to the result generated by the given work. AREA 2014 also indicated that the European DC market share would be between 2 and 3 TWh/y [61]. Overall, the aforementioned sources confirmed that the European useful energy demand for DC reached a share of approximately 1 to 2% of the European cooling market energy demand. It is important to state that no sources could be found indicating that the DC sales in Europe were higher than those calculated in the current study.

It has been observed that the European DC market is expected to considerably grow within the next decade, reaching a value of 12 TWh/y in 2030 [19]. However, the given indication appears to be questionable because the growth of the investigated market appears to be limited in the past decade. As already mentioned, Dalin et al. provided a value of 2.3 TWh/y in 2003 [13]. Tvärne et al. confirmed the latter-mentioned value and also indicated respective values for 2007 and 2009, which ranged from 2 to 3 TWh/y [19].

The penetration of DC varied considerably from country to country. The EU Nordic countries showed a high capacity installed compared with other European nations. Around 80% of the DC capacity of Europe was found in Sweden, Finland, and Denmark; Sweden alone accounted for almost 75%. In contrast, there were a number of countries located in Warm Europe that appeared to not make use of a DCS. Moreover, the latter could make use of the sea as a temperate cold source.

District cooling in the EU is characterized by a high growth potential. The primary potential market for DC concerns service sector buildings with commercial areas for offices and retail. Campus areas, hospitals, airports, and universities can also serve as a local hub for DC applications. Depending on the location and distance to city areas, campus areas can be connected to a larger DC system. Moreover, there is potential for DC in residential areas that have flats in Central and Southern Europe [13,56,57]. According to the European energy targets presented in the revised Renewable Energy Directive (REDII), the RES applications should be improved throughout the whole EU27 MSs and the implementation of RESs such as biomass, solar-thermal, geothermal, solar-photovoltaic, and waste heat in the DC sector could enhance the reach of the energy goals set by the REDII.

Lastly, it is worth mentioning that little information was found regarding the DC sector during the literature review phase. Overall, the whole cooling sector is barely explored and, therefore, retrieving relevant information difficulties have been experienced throughout this study, especially regarding accessibility. It is worth pointing out that data/information retrieval from private companies might encounter issues due to privacy policies [62]. However, the accessibility for research and development (R&D) reasons of the latter is auspicial and even beneficial for the companies themselves.

## 5. Conclusions

This study presented a collection and comparison of data, offering insights into the district cooling market within the European Union plus the United Kingdom in 2016. In addition, the study provided a statistical elaboration and calculation of the data. The main results of the study were:

- District cooling accounted for a small part of the useful cooling demand of Europe; around 3 TWh/y. The latter was equal to around 1 to 2% of the entire cooling market of Europe.
- The presence of district cooling systems varied considerably from country to country. It was discovered that throughout the European Union, around eight countries appeared to not make use of this cooling technology.
- It was observed that the European Scandinavian countries were characterized by a high amount of installed capacity when compared with the remaining European MSs. About 80% of the entire European Union district cooling capacity could be found in Sweden, Finland, and Denmark. Notably, district cooling systems installed in Sweden accounted for almost 75%.
- In contrast with the results presented in the previous bullet point, there were a series of MSs mainly located in Southern Europe for which no presence of district cooling systems could be found.
- Overall, it could be stated that there was a slight but continuous growth in Europe for decades of district cooling installations with a high growth potential, especially with the synergies for renewable energy sources. The major potential market for the growth of the district cooling systems was the service sector.

Although the European district heating market is well-investigated in scientific literature, the district cooling branch presents a major lack of information. The study team experienced notable difficulties in finding data/information regarding the analyzed field. Moreover, an amount of the data/information regarding the treated topic was not freely accessible, was outdated, was mixed in different repositories, or was fragmented.

To conclude, it would be interesting to discover to which extent district cooling plants in Europe are powered by renewable energy sources and which those are. Additionally, future works may focus on analyzing in more detail the impact of specific input data such as seasonal energy efficiency ratio values, equivalent full-load hours, and the free cooling portion to further validate these data.

**Author Contributions:** Conceptualization, S.P., P.R. and L.K.; data curation, S.P. and A.Z.; writing, S.P., G.Q. and A.Z.; supervision, L.K. and P.R.; validation, P.R., L.K., A.N., L.B., M.H. and E.W. All authors have read and agreed to the published version of the manuscript.

**Funding:** The Directorate-General financed this research for the European Commission/Energy, service contract number ENER/C1/2018-493/SI2.813961.

**Institutional Review Board Statement:** Not applicable.

**Informed Consent Statement:** Not applicable.

**Data Availability Statement:** Not applicable.

**Acknowledgments:** The authors thank the Department of Innovation and Research, and the University of the Autonomous Province of Bozen/Bolzano for covering the open access publication costs.

**Conflicts of Interest:** The information and views set out in this article are those of the authors and do not necessarily reflect the official opinion of the European Commission.

**Abbreviations**

| Acronym | Name |
|---------|------|
| AC | Air conditioning |
| ASHRAE | American Society of Heating, Refrigerating And Air Conditioning Engineers |
| DC | District cooling |
| DCS | District cooling systems |
| DHW | Domestic hot water |
| EEA | European Environment Agency |
| EED | Energy Efficiency Directive |
| EFLHs | Equivalent full-load hours |
| Eq | Equation |
| EU | European Union |
| FEC | Final energy consumption |
| GHG | Greenhouse gas |
| H&C | Heating and cooling |
| H2020 | Horizon 2020 |
| PC | Process cooling |
| R&D | Research and development |
| RED | Renewable energy directive |
| RES | Renewable energy sources |
| SC | Space cooling |
| SEER | Seasonal energy efficiency ratio |
| TDHP | Thermally driven heat pump |
| UED | Useful energy demand |
| VC | Vapor compression |

**Nomenclature**

| Acronym | Name |
|---------|------|
| Equivalent full-load hours | h |
| Percentage | % |
| Wh | Watt/hour |

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
