# Peer review of "Recent Advances in District Cooling Diffusion in the EU27+UK: An Assessment of the Market"

_sustainability, doi:10.3390/su14074128_

Round 1

Reviewer 1 Report

The reviewed manuscript is at a good level in terms of the originality of the presented solutions and formulated final conclusions and will probably meet with the interest of readers, and its results present a real picture of the District Cooling sector in the EU27+ UK countries. The manuscript presents a detailed overview of DC systems in recent decades with a focus on data for 2016. The weaker point of the manuscript is the lack of this analysis for recent years, but this is due to the lack of reliable data for this period.

The manuscript has both a clear descriptive form, and statistical studies of a relatively simple form are clearly presented in the form of tables and graphs.

However, the presented manuscript requires some minor changes, primarily to improve its readability, i.e. additions and corrections should be made to the presented text.

The following are selected detailed remarks:

- it is recommended to introduce a list of abbreviations, before Chapter 1. Introduction,

- it is recommended to introduce a Nomenclature with units for individual quantities, after Chapter 5. Conclusions,

- lines153-155, subsections cannot be divided into sections, please correct according to the names of the chapters in the manuscript,

- line 194, the note probably applies to absorption and adsorption equipment, so please add the word adsorption or replace the word absorption with the word sorption.

Author Response

Please see the file attached.

Reviewer 2 Report

The manuscript can be slightly improved. There are few typos and language flaws. The methodology should be clearly elaborated and explained. 

Author Response

Please see the file attached.

Reviewer 3 Report

The manuscript entitled “Recent Advances of District Cooling Diffusion in the EU27+UK: Assessment of the Market" focuses on data and information from the past decade and draws a picture for the year 2016 accordingly. This is an interesting study and the authors have collected a unique dataset. Minor shortcomings with the manuscript are as follows.

·        Abstract could be improved. In particular the middle two sentences. More generally, I suggest to focus the results and discussion. Numerical key findings need to mention into the abstract.

·        lierature review can be improved. The connection between the research gap and targeted objectives can be more prominent 

·        In the methodology all parameters i.e., EFLHs SEER should be further elaborated and refined for the clarity of the general readers. 

·        In data collection several sites are mentioned that provides the information regarding DC so why the study is being caried out for the market in EU27 + UK. Furthermore, discuss the impact of your study. 

·        It would be better is possible to add the factors that enhance the cooling requirement. 

·        How about adding the cost analysis to make the study more effective? Only if it is easy for authors

·        In results the Figure 1 shows the number of DC units as per applications in the different districts. I will suggest to add the legends and the range of the longitude and latitude in QGIS. 

·        It would be better to not mention the countries whose data is not available instead of mentioning ‘/’.

·        Kindly elaborate the free cooling space (validator). 

·        Kindly inprove the section 4. Make it comprehensive. 

·        Revise the conclusion. Make it concise and only necessary information must be discussed

Author Response

Please see the file attached.
